# FSH-Induced Nuclear Exclusion of FOXO1 Mediated by PI3K/Akt Signaling Pathway in Granulosa Cells Is Associated with Follicle Selection and Growth of the Hen Ovary

**DOI:** 10.3390/cells14231864

**Published:** 2025-11-26

**Authors:** Chunchi Yan, Yu Ou, Xue Sun, Yuhan Sun, Jinghua Zhao, Ning Qin, Rifu Xu

**Affiliations:** 1College of Animal Science and Technology, Jilin Agricultural University, Changchun 130118, China; 2College of Animal Husbandry and Veterinary, Jinzhou Medical University, Jinzhou 121004, China

**Keywords:** PI3K/Akt pathway, FOXO1, nuclear exclusion, hen ovary, follicle selection

## Abstract

Follicle selection is a pivotal process that determines which dominant prehierarchical follicle will enter the preovulatory hierarchy in the hen ovary and directly affects egg-laying productivity, in which granulosa cells (GCs) are characterized by active proliferation and significantly enhanced *FSHR* mRNA expression. Increasing evidence has shown that the PI3K/Akt signaling pathway and its important target and effector FOXO1, which promotes GC apoptosis, play crucial roles in ovarian follicular development in mammals. To investigate the molecular mechanism by which follicle-stimulating hormone (FSH)-mediated forkhead box O1 (FOXO1) participates in follicle selection, we treated granulosa cells from 6–8 mm prehierarchical follicles of chickens with FSH and leptomycin B (LMB). The results showed that under FSH and/or LMB treatment, the expression levels of FSHR, FOXO1, and its phosphorylated forms (p-FOXO1) at the predicted protein kinase B (PKB/Akt) phosphorylation sites Thr^24^, Ser^248^, and Ser^311^ were differentially regulated. The subcellular localization of p-FOXO1 in hen ovarian GCs was determined by Western blotting and immunofluorescence staining (IF) analysis. And the expression of FOXO1 was significantly reduced, whereas the expression of p-FOXO1 corresponding to the PKB phosphorylation sites Ser^248^ and Ser^311^ was noticeably boosted in cultured GCs induced by FSH, accompanied by exclusion of FOXO1 from the nucleus to the cytoplasm. Subsequently, the effects of the PI3K/Akt signaling pathway on phosphorylation levels and nuclear exclusion of p-FOXO1 at the sites Ser^248^ and Ser^311^ were examined. The results indicate that the PI3K/Akt-dependent phosphorylation at these sites directly resulted in nuclear exclusion of FOXO1 in ovarian GCs, in which the Ser^248^ site is more essential than the Ser^311^ site. Subsequently, the FSH-induced acetylation of FOXO1 mediated by the cAMP/PKA pathway can enhance the phosphorylation level of FOXO1 at the Ser^248^ site. In summary, our findings demonstrate that FSH induces FOXO1 phosphorylation, nuclear exclusion, and functional inactivation by activating the PI3K/Akt signaling pathway. Moreover, during follicular development and selection, FOXO1 acts as a pivotal mediator linking the PI3K/Akt and P62/Keap1/Nrf2 signaling pathways to regulate granulosa cell proliferation and apoptosis, thereby exerting a central regulatory role.

## 1. Introduction

Egg production is a desirable economic trait in laying hens, which greatly depends upon ovarian follicle development, selection, and a well-organized preovulatory hierarchy [1,2,3]. In the hen ovary, a single undifferentiated follicle per day is selected into the preovulatory hierarchy on an approximate daily basis [4], usually occurring from a small cohort of prehierarchal follicles sized 6 to 8 mm in diameter, and this initiates rapid growth and final maturation before ovulation [1,5]. Follicle selection is driven by the functional differentiation of granulosa cells (GCs) under the precise regulation of follicle-stimulating hormone (FSH) and its receptor (FSHR) [6,7]. In the selected follicles, elevated *FSHR* mRNA expression [8] and cyclic adenosine monophosphate (cAMP) levels activate the downstream cAMP/PKA signaling cascade [9,10,11]. Beyond this classical pathway, additional signaling routes, including PI3K/Akt and p62/Keap1/Nrf2, also contribute to the regulation of follicular growth and differentiation [12,13,14].

FSH is a pituitary glycoprotein hormone playing a pivotal role in the growth and differentiation of ovarian follicles by stimulating GC proliferation and differentiation through the PI3K/Akt kinase pathway and the cAMP/PKA pathway [12,15,16]. The FoxO transcription factor family, as one of the most important downstream targets of the PI3K/Akt pathway, is emerging as a central point in an array of cellular functions, including cell proliferation, differentiation, and apoptosis [13,17,18]. The main mammalian FOXO family members are forkhead box O1 (FOXO1, also known as FKHR), FOXO3, FOXO4, and FOXO6 [19,20]. Due to their important roles in cell development, differentiation, apoptosis, cell cycle arrest, mitophagy, cellular homeostasis, and follicular atresia, FOXO has quickly become a very topical area of life science research [18,21]. However, the role of FOXO1 in ovarian follicle development and function has not been fully established in chickens. 

The FOXO transcription factors are highly expressed in ovarian granulosa cells [22,23,24]. Accumulating evidence has suggested that the FOXO members are involved in follicle development, growth, and atresia [22,25]. FOXO activity is regulated by numerous post-translational modifications [13]. Phosphorylation of FOXOs induced by the FSH-PI3K/Akt signaling pathway results in the exclusion of the FOXO factors from the nucleus to the cytosol, and inhibiting FOXO-dependent transcription [26,27]. As a proapoptotic molecule that belongs to the FOXO subfamily, FOXO1 plays a critical role in promoting follicular atresia and granulosa cell apoptosis of the mammalian ovaries [24,28,29]. Oppositely, FSH enhances the expression of genes involved in cell proliferation and estrogen production, promotes follicular growth, and decreases granulosa cell apoptosis [26,30]. As a result, FOXO1 activity is negatively influenced by FSH in the GCs [26]. Furthermore, the investigation of the regulatory mechanism of FoxO1 response to FSH-mediated prevention of GC apoptosis has shown that FSH induces protein kinase B (PKB; also known as Akt) phosphorylation of FoxO1 at three predicted amino acid residues (Thr^24^, Ser^256^ and Ser^319^) in human follicular GCs, and triggers nuclear translocation of FoxO1 by activation of the PI3K/Akt signaling pathway [13,27,31,32]. In contrast, suppression of the PI3K/Akt pathway results in dephosphorylation and nuclear retro-translocation of FoxO1, which induces cell cycle arrest and apoptosis in mice [13,29]. Our previous study demonstrated that FSH promotes the phosphorylation and nuclear exclusion of FOXO3/4 proteins, leading to their retention in the cytoplasm and the loss of transcriptional activity, thereby suppressing the pro-apoptotic effects of FOXO3/4 in chicken granulosa cells. However, the molecular regulatory mechanism of FOXO1 in chickens remains unclear. Although it has been reported that the phosphorylation of FoxO1 contributes to its nuclear exclusion depends on the fuction of nuclear export proteins [33,34,35,36] whether FSH regulates the phosphorylation and nuclear translocation of FOXO1 in chicken granulosa cells via the PI3K/Akt signaling pathway, and how this process influences follicle selection and growth, has not yet been elucidated. Additionally, there may be functional crosstalk between the PI3K/Akt pathway and the oxidative stress-related p62/Keap1/Nrf2 signaling pathway. The latest reports demonstrated that the P62/Keap1/Nrf2 pathway is implicated in the regulation of mammalian ovarian granulosa cell apoptosis and ferroptosis [37,38,39]. Even so, the precise role and response mechanism of FOXO1 to FSH-induced activities of the PI3K/Akt pathway, the cAMP/PKA and P62/KEAP1/Nrf2 pathways in follicle growth, atresia, and follicle selection of hen ovary remain poorly understood nowadays.

The present study aimed to reveal the biological effect and molecular mechanisms of FSH-induced FOXO1 phosphorylation and its nuclear exclusion in vitro on the follicle selection and growth of hen ovary via the PI3K/Akt signaling pathway. Our results demonstrated that FSH-induced nuclear exclusion of FOXO1 is closely associated with hen ovarian follicle selection and growth by promoting higher *FSHR* mRNA expression and GC proliferation and inhibiting cell apoptosis via the PI3K/Akt signaling pathway. Furthermore, PI3K/Akt-dependent phosphorylation at the two sites, Ser^248^ and Ser^311^, results in nuclear exclusion of FOXO1, in which the activated RAN, CRM1, and 14-3-3 are implicated as the nucleocytoplasmic transport factors. Further study proved that the FOXO1 factor plays a pivotal role in ovarian follicle growth and selection by bridging the crosstalk between the PI3K/Akt and P62/Keap1/Nrf2 signaling pathways in chickens.

## 2. Materials and Methods

### 2.1. Ethics Approval

In this study, all procedures conducted in chickens were approved by the Institutional Animal Care and Use Committee (IACUC) of Jilin Agricultural University (Changchun, China). The animal experiment was performed in compliance with the ARRIVE guidelines [6]. Chickens were sacrificed before removing organs following the IACUC guidelines for experimental animals (Permission No. GR (J) 19-030). The ethical approval date is 1 April 2023. Euthanasia of the hens was fully compliant with the Chinese applicable laws and regulations concerning the care and use of laboratory animals, which were issued based on the Regulations for the Administration Affairs Concerning Experimental Animals of the State Council of the People’s Republic of China (2017 Revision). All efforts have been made to minimize the suffering of the animals.

### 2.2. Animal and Granulosa Cell Culture and Treatment with Reagents

A commercial strain of Lohmann Brown laying hens was raised in laying batteries according to the standard husbandry practices, as previously reported [6]. Eighty Lohmann Brown laying hens from the breeding base of Jilin Agricultural University were used in this study. The hens were kept in single cages, freely accessing feed, water, and exposed to a 16L:8D photoperiod. At 21 weeks, 22 hens were procured from the population and euthanized, and the prehierarchical follicles sized 6–8 mm in diameter were immediately collected. The ovarian GCs were subsequently isolated from the follicles and cultured according to a protocol that was previously reported by us [3]. A cell suspension was then prepared from the isolated GCs before adding M199 culture medium (Biosharp Life Sciences, Hefei, China) containing 10% newborn calf serum, 100 IU/mL of penicillin, 75 IU/mL of streptomycin, and 10 ng/mL of insulin. The GCs were then incubated in 5% CO_2_ and 95% air at 37 °C. Subsequent experiments were performed after 12 h when cell adhesion had occurred. Cells were set up as follows: control (PBS, Procell, Wuhan, China); FSH (Follicle-stimulating hormone, 10 ng/mL, 12 h; Selleck Chemicals, Houston, TX, USA); LMB (Leptomycin B, 10 ng/mL, 12 h; Targetmol, Boston, MA, USA) [40]; Ly294002 (Ly294002, 25 μM, 2 h; Targetmol, Boston, MA, USA) [41]; 740-Y-P (740-Y-P, 20 μM, 24 h; Houston, TX, USA) [42]; KH7 (KH7, 25 μM, 1 h; Targetmol, Boston, MA, USA) [43]; TSA (Trichostatin A, 1 μM, 2 h; Targetmol, Boston, MA, USA). After the incubation, the culture medium and whole-cell extracts were obtained for the subsequent analyses.

### 2.3. Construction of Recombinant Plasmids and Cell Transfection

After the chicken FOXO1 cDNA sequence (GenBank accession: NM_204328.2) was amplified from a chicken cDNA library by PCR, the FOXO1 wild-type (WT), FOXO1 Ser-248A, FOXO1 Ser311A, and FOXO1 mutation-type (MT) were cloned into the pcDNA3,1(+) plasmid (Integrated Biotech Solutions, Shanghai, China) by using the specific primers and then subcloned into a pcDNA3,1(+) expression vector to generate the pcDNA3,1(+)-FOXO1 WT, FOXO1 Ser-248A, FOXO1 Ser-311A, FOXO1 MT expression construct. Briefly, the GCs were transfected with the plasmid pcDNA3,1(+)-FOXO1, FOXO1 Ser-248A, FOXO1 Ser311A, FOXO1 MT and pcDNA3,1(+) blank vector using Lipofectamine 2000 (Biosharp, bl632b, Hefei, China). Cells were cultured in Opti-MEM (hexadimethrine bromide, Sigma) and incubated at 37 °C with 5% CO_2_. After 6 h of culture, the GCs were cultured in complete medium for 24 h and then lysed for immunoblot analysis and RT-qPCR analysis.

### 2.4. Transfection of siRNA

Specific siRNAs targeting the *P62* gene were designed using an InvivoGen siRNA Wizard v3.1. All designed siRNA sequences were blasted against the chicken genome database to eliminate the cross-silence phenomenon with non-target genes. A most effective LATS2-specific siRNA was further screened by RT-qPCR: 5′-GAUACAAGACAUGGUGAUAGA-3′. Scrambled siRNA was used as the negative control siRNA: 5′-UAUCACCAUGUCUUGUAUCUG-3′. As mentioned above, GCs were plated in 6-well plates, and the siRNAs were transfected into the cultured cells using Lipofectamine 2000 (Biosharp, Hefei, China) according to the manufacturer’s instructions, as we previously reported [6].

### 2.5. Immunofluorescence Assay

In the process of cell culture, six-well tissue culture plates were filled with circular glass coverslips, which were placed there to promote cell growth. Following this, cells were immobilized by use of 4% paraformaldehyde (PFA, Beyotime, Shanghai, China), and the ends of the coverslips were homogeneously coated with an immunostaining guard pen (Biosharp Life Sciences, Hefei, China) to generate a 1 mm oily barrier. The cells were then incubated with primary antibodies: anti-FOXO1 (Proteintech, Wuhan, China, 1:500 dilution) at 4 °C for 12 h. After washing with PBS, the cells were incubated with fluorescent-conjugated goat anti-rabbit IgG antibody (Epizyme Biotech, Shanghai, China, 1:2000 dilution). Nuclei were stained with 4′,6-diamidino-2-phenylindole (DAPI, Epizyme Biotech, Shanghai, China). All immunofluorescence images were captured by laser confocal microscope (Zeiss, Oberkochen, Germany).

### 2.6. Western Blotting Assay

The procedure made use of an RIPA lysis buffer (Biosharp Life Sciences, Hefei, China) to treat the cells for the purpose of protein extraction from the samples. This was then followed by the addition of a protease inhibitor cocktail (Epizyme Biotech, Shanghai, China) and phosphatase inhibitor cocktail (Epizyme Biotech, Shanghai, China), with subsequent sonication for fragmentation. The concentration of the protein was then adjusted following the instructions provided in the BCA protein assay kit (Beyotime, Shanghai, China). Then, protein samples were added with 5× loading buffer (Yamay Bio, Shaihai, China, LT101) and heated at 95 °C for 5 min. Nitrocellulose membrane transfer was then completed after electrophoresis at 150 V in a 10% SDS PAGE gel (Epizyme Biotech, Shanghai, China). After blocking with 5% skim milk, the membrane was incubated overnight at 4 °C with 1:1000 diluted primary antibody, as shown in Table 1. After washing with PBS, the samples were incubated with a horseradish peroxidase-conjugated goat anti-mouse IgG antibody diluted 1:1000 as a secondary antibody for 1.5 h. The IgGs are shown in Table 2. Images were taken and visualized by an ECL Plus Western blotting detection system according to the manufacturer’s instructions.

### 2.7. Quantitative Real-Time PCR Analysis

To assess the mRNA expression of the target genes in the GCs, real-time quantitative reverse transcriptase PCR (qRT-PCR) was conducted according to the previously described method. The cDNA sequence of P62 was referenced to the literature. The primer sequences used in this study are shown in Table 2. The reaction system contained SYBR (Biosharp, Anhui, China) 10 μL, 0.4 μL upper- and 0.4 μL lower-primer, and cDNA 2 μL. The amplification conditions used were 95 °C for 10 min and 95 °C for 15 s, with an annealing temperature of 60 °C for 60 s and a total of 40 cycles. The 18S rRNA was used as a normalizing gene.

### 2.8. Flow Cytometry Assay

The cells processed as explained earlier were digested with trypsin without EDTA, and the cells were collected. After centrifugation, the supernatant was removed, and the cells were washed with PBS, suspended again, and re-centrifuged to eradicate the supernatant. The annexin V/PI double staining kit (MULTISCIENCES, Hangzhou, China) was used to detect the level of apoptosis of granulosa cells in different treatment groups. The prepared cell samples were stained according to the instructions of the annexin V-FITC/PI apoptosis test kit, which was instantly analyzed using the CytoFlex flow cytometer(Becton, Dickinson and Company, Franklin, TN, USA). In accordance with the scatter plot of the bivariate flow cystoscope, the proliferating live cells were displayed as (FITC-/PI-) in the lower left quadrant. Necrotic cells, or non-viable cells, were displayed in the upper right quadrant as (FITC+/PI+). Some believed these cells to be non-viable apoptotic cells, or dead cells, which were represented as (FITC-/PI+). Apoptotic cells were exhibited in the lower right quadrant as (FITC+/PI-). Each experiment was performed in triplicate and repeated five times. The relative positive percentage was calculated as the average of five group values.

### 2.9. Co-Immunoprecipitation Assay

In this study, a co-immunoprecipitation assay was performed according to the method previously published by us [6]. Briefly, the buffer was freshly prepared, containing 8.766 g/L NaCl and 2.8392 g/L Na_2_HPO4. Then, protein A/G was washed with the buffer. The cellular extracts (1 mg/μL, 10 μL) were incubated with the antibody against FOXO1 and protein A/G beads according to the kit instructions (rProteinA/G Beads 4FF) (Solarbio Science & Technology CO., Ltd., Beijing, China). The eluted samples or cell lysates were added to 5×SDS loading buffer (Yamay Bio, China, LT101) and heated at 95 °C for 5 min to denature the proteins. Subsequently, the immunocomplexes were separated by SDS-polyacrylamide gel electrophoresis and transferred to PVDF membranes. The membranes were then incubated with the primary antibody against 14-3-3 (Proteintech, Wuhan, China, 1:1000 dilution)/CRM1 (Proteintech, Wuhan, China, 1:1000 dilution)/RAN (Proteintech, Wuhan, China, 1:1000 dilution), followed by the secondary antibody. Immunoreactive bands were detected using Chemistar ECL Western Blotting Substrate (Tanon Science & Technology Co., Ltd., Shanghai, China).

### 2.10. Statistical Analysis

Statistical analysis was implemented using the SPSS12.0 software package. All the experiments were repeated at least three times using different batches of sampled chicken. The data were analyzed with a one-way ANOVA and Tukey’s multiple-comparison test when more than two groups were involved or using a Student’s *t*-test when treatment and control groups were compared after confirming normal distributions for parametric analysis by the Shapiro–Wilk test. *p* < 0.01 or *p* < 0.05 was considered to be statistically significant. All experiments in this article included 3 replicates per group.

## 3. Results

### 3.1. Influence of FSH on Phosphorylation of FOXO1 in Hen Ovarian GCs

To estimate the phosphorylation sites of FOXO1, alignment of the amino acid sequence of chicken FOXO1 protein (accession number NP_989659 in the GenBank database) with mammals and other poultry was conducted. It was found that the sequence of chicken FOXO1 was highly conserved, being 71.35%, 75.84%, and 75.37% identities to human, mouse, and cow FoxO1, and 92.57% and 91.75% homologous with goose and duck FOXO1 at the amino acid sequence level, respectively. As expected, three predicted protein kinase B (PKB/Akt) phosphorylation sites (Thr^24^, Ser^248^, and Ser^311^) of chicken FOXO1 were completely conserved, corresponding to the three residues, Thr^24^, Ser^256^, and Ser^319^, of human FoxO1, and corresponding to mouse (Thr^24^, Ser^253^, and Ser^316^), duck (Thr^24^, Ser^277^, and Ser^340^), and goose (Thr^24^, Ser^272^, and Ser^335^) at the amino acid level (Appendix A).

To determine the influence of FSH and the exportin1 inhibitor leptomycin (LMB) on the phosphorylation and cellular localization of FOXO1 in hen ovarian GCs, the expression levels of FSHR, FOXO1, and phosphorylated FOXO1 (p-FOXO1) proteins in the GCs were examined by Western blotting (Figure 1a–f). The expression levels of FSHR were significantly increased in the GCs induced by 10 ng/mL FSH (*p* < 0.01), but no different expression levels were observed between the FSH and LMB (10 ng/mL) co-treatment group with the FSH treatment (*p* > 0.05, Figure 1b). However, the expression level of total FOXO1 was markedly reduced by FSH only (*p* < 0.01), and expectedly, the decreased expression level was abolished by LMB treatment (*p* > 0.05, Figure 1c). Although the levels of p-FOXO1 corresponding to the PKB phosphorylation site Thr^24^ remained unchangeable in the cells induced by FSH or by FSH and LMB (*p* > 0.05, Figure 1d), the expression levels of p-FOXO1 corresponding to the sites Ser^248^ and Ser^311^ were noticeably boosted, and higher expression levels of p-FOXO1 corresponding to the two sites were observed by co-treatment with FSH and LMB than with FSH alone (*p* < 0.05, Figure 1e,f). Furthermore, the treatment with FSH alone leads to the exclusion of FOXO1 protein from the nucleus to the cytoplasm of the GCs. However, in the treatment with FSH+LMB, FOXO1 was observed to be largely located in the nucleus (Figure 1g). These results indicated that the FSH-induced phosphorylation of FOXO1 in hen ovarian GCs may be mainly attributed to the two sites, Ser^248^ and Ser^311^. Meanwhile, the phosphorylation of FOXO1 at the sites may be closely associated with enhanced nuclear exclusion of the p-FOXO1 protein.

### 3.2. FSH-Induced FOXO1 Phosphorylation Mediated by FSHR/PI3K/Akt Signaling Pathway

Previous studies in mammals have indicated that FoxO1 is involved in GC apoptosis induced by FSH via mediation of the PI3K/Akt pathway [13,27]. To explore the precise effects of the activated FSHR/PI3K/Akt signaling pathway on the FSH-induced phosphorylation of FOXO1 in chicken follicular GCs, FSH (10 ng/mL) was administered to the medium of the GC culture for 12 h with or without Ly294002 (25 μM), a PI3K inhibitor, to prevent the PI3K/Akt pathway. The results showed that the protein levels of FSHR increased significantly in the cells cultured with FSH in the absence or presence of Ly294002 (*p* < 0.05, Figure 2a,b). Furthermore, p-Akt expression rose noticeably in the absence of the PI3K inhibitor (*p* < 0.05, Figure 2c), and synchronously, the expression levels of p-FOXO1 corresponding to the phosphorylation sites Ser^248^ and Ser^311^ were sharply elevated in the cells with treatment of FSH alone (*p* < 0.05, Figure 2d–f). Nevertheless, the increased expression level of the p-Akt induced by FSH was completely abolished in the presence of Ly294002 (Figure 2c). Subsequently, the elevated expression level of p-FOXO1 at the two sites stimulated by FSH was totally abrogated in the presence of Ly294002 or Ly294002 and LMB (*p* > 0.05, Figure 2e,f), respectively. And the total FOXO1 expression upon FSH treatment was significantly reduced as compared with those treated simultaneously by Ly294002 or Ly294002 and LMB (*p* < 0.05, Figure 2g). These results strongly supported that the FSH-induced phosphorylation of FOXO1 in ovarian GCs was predominantly determined by the two sites, Ser^248^ and Ser^311^, and mediated by the FSHR/PI3K/Akt signaling pathway. Additionally, the phosphorylation of FOXO1 may directly lead to its nuclear exclusion, which caused the reduction of FOXO1 expression by the enhanced degradation of the p-FOXO1 protein in the cytoplasm of the GCs.

To explore the influence of cAMP/PKA signaling on the FSH-induced FOXO1 phosphorylation mediated by the PI3K/Akt pathway, KH7, a cAMP inhibitor, was utilized to block the activity of the cAMP/PKA pathway. The results demonstrated that under treatment with FSH, the expression levels of PKACA and acetylated FOXO1 (acFOXO1) were markedly boosted compared with the control group (*p* < 0.05). However, the elevated expression abundance of PKACA and acFOXO1 proteins in the cells was completely abolished by the administration of KH7 (*p* > 0.05, Figure 2h–j). Subsequently, Trichostatin A (TSA), a histone deacetylase inhibitor [44,45], and Ly294002 were used to treat the cells to detect the effect of FOXO1 acetylation on FOXO1 phosphorylation. The results showed that under TSA treatment alone, the phosphorylation level of FOXO1 at the Ser^248^ site was sharply elevated (*p* < 0.01, Figure 2k,l), whereas the level of p-FOXO1 at the Ser311 site was not markedly different from that of the control group. In the case of Ly294002 treatment, there was no significant difference in the expression levels of p-FOXO1 at the Ser^248^ and Ser^311^ sites compared with the control group (Figure 2l,m). In the co-treatment with Ly294002 and TSA, there was also no significant influence on the expression levels of p-FOXO1 at the two sites compared with the control or with the treatment with Ly294002 alone (Figure 2l,m). The results indicated that the FSH-induced acetylation of FOXO1 mediated by the cAMP/PKA pathway can prompt the phosphorylation level of FOXO1 at the Ser^248^ site in chicken GCs.

### 3.3. Roles of the FOXO1 Phosphorylation Sites in Regulating Its Nuclear Exclusion

To clarify the roles of the FOXO1 phosphorylation sites (Ser^248^ and Ser^311^) in regulating its exclusion from nucleus to cytoplasm, overexpression vectors for wild-type, mutation-type S248A (replacing serine with alanine at the 248 site), S311A (replacing Ser^311^ with Ala^311^), and S248A/S311A FOXO1 were constructed to transfect the ovarian GCs, and 740-Y-P, a cell-permeable PI3K activator [46], was simultaneously administered in the transfected cells. It was found that the expression levels of p-FOXO1 at the Ser^248^ site in the wild-type and the mutation-type S311A groups were significantly elevated compared with the mutation-type S248A alone or the S248A&S311A FOXO1 expression vector, respectively (*p* < 0.05, Figure 3a,b). And the expression levels of p-FOXO1 at the Ser^311^ site were remarkably boosted in the wild-type group compared with the groups transfected with the mutant S248A, S311A, or S248A&S311A FOXO1 vector, respectively (*p* < 0.05, Figure 3a,c). These data proved that the two sites, Ser^248^ and Ser^311^ on FOXO1, were the key phosphorylation targets of the PI3K/Akt signaling pathway, and the substitution occurring at the FOXO1 Ser^311^ residue has no influence on the expression of p-FOXO1 at the Ser^248^ site, but the mutation of the FOXO1 Ser^248^ residue prevents the expression of p-FOXO1 at the Ser^311^ site.

To subsequently assess the impacts of phosphorylation at two sites on the subcellular localization of FOXO1 in the GC cells, immunofluorescence staining analysis was implemented under induction of the activated PI3K/Akt pathway. It was shown that the wild-type FOXO1 was distributed largely in the extra-nuclear compartment of the cultured GCs under treatment with 740-Y-P (Figure 3d,e), but the majority of FOXO1 with the Ala^248^ residue (replacing the Ser^248^) was apparently emerged within the nuclear compartment (Figure 3f). Furthermore, as overexpression of FOXO1 with the mutant Ala^311^ residue (replacing the Ser^311^), the increased p-FOXO1 was mainly caused by phosphorylation at the Ser^248^ and contributed to the location of FOXO1 in the extra-nuclear compartment since the FOXO1 with the mutant Ala^311^ site did not suffer from phosphorylation at the Ser^311^ site of FOXO1 (Figure 3g). Nevertheless, when overexpression of FOXO1 with the mutant Ala^248^ and Ala^311^ residues (replacing the Ser^248^ and Ser^311^), the majority of FOXO1 with the Ala^248^ and Ala^311^ residues was mainly localized in the nuclear compartment that resulted from the mutation at the Ser^248^ and Ser^311^ sites of FOXO1, preventing it from phosphorylation induced by the activated PI3K/Akt pathway (Figure 3h). The current findings indicate that PI3K/Akt-dependent phosphorylation at these sites resulted in the nuclear exclusion of FOXO1 in ovarian GCs, in which the Ser^248^ site plays a more essential role than the Ser^311^ site.

As shown in Figure 3, when wild-type FOXO1 was phosphorylated in cultured GCs with 740-Y-P treatment, its immediate interaction with the 14-3-3 protein was strengthened, whereas the binding of the 14-3-3 protein to FOXO1 with the mutant Ala^248^ residue (un-phosphorylated) significantly decreased. Under the same treatment, on the other hand, the binding of the 14-3-3 protein to FOXO1 with the mutant Ala^311^ residue (un-phosphorylated) kept consistent with the wild-type (Figure 4a,b), further supporting that the Ser^248^ residue directly contributed to the increase in the FOXO1 nuclear exclusion induced by the PI3K/Akt-dependent phosphorylation at this site, and its phosphorylation was indispensable for the combination of the 14-3-3 protein with the FOXO1 factor. Under similar treatment conditions with 740-Y-P, unexpectedly, the binding levels of CRM1 protein to FOXO1 with a mutant Ala^248^ or/and Ala^311^ residue remained unchanged (Figure 4c,d), indicating the combination of CRM1 with FOXO1 was not affected by FOXO1 phosphorylation via the PI3K/Akt pathway. Furthermore, under induction of the overexpressed RanQ69L mutant (replacing Glutamine-69 with Leucine) that can bind GTP (in RanGTP or GTPase FOXO1 form) and was added to the cytosol [47], the binding levels of CRM1 protein to FOXO1 were noticeably increased in the cells treated with/without Ly294002, while the combination levels of the 14-3-3 protein to FOXO1 significantly attenuated in the presence of Ly294002 (Figure 4e–g). The results demonstrated that the activated Ran enhanced the combination of FOXO1 and CRM1, which was not affected by the influence of the PI3K/Akt-dependent phosphorylation of FOXO1. Conversely, the combination of FOXO1 and 14-3-3 mainly depended on the PI3K/Akt-dependent phosphorylation of FOXO1 but was not affected by the Ran activity. Moreover, regardless of treatment with 740-Y-P or Ly294002, the binding levels of Ran protein to FOXO1 with a mutant Ala^248^ or Ala^311^ residue obviously diminished (*p* < 0.01, Figure 4h–j), which shows that the two PKB phosphorylation sites of FOXO1 are essential for its combination with Ran.

Collectively, our findings demonstrated that activated Ran promoted the combination of FOXO1 and CRM1 in a PI3K/Akt signaling-independent manner, but the two PKB phosphorylation sites in FOXO1 are indispensable for its combination with Ran. On the contrary, the combination of FOXO1 and 14-3-3 mainly depended on FOXO1 phosphorylation induced by the PI3K/Akt signaling pathway but was not affected by Ran activity.

### 3.4. Effects of FSH-Induced FOXO1 Nuclear Exclusion on Ovarian Follicle Growth and Selection Mediated by PI3K/Akt Signaling Pathway

As shown in Figure 5, double immunofluorescence staining demonstrated that the FOXO1 protein was localized in the GC nucleus and cytoplasm in the control group, whereas the nuclear FOXO1 almost failed to be visualized in nuclei but mainly accumulated in the cytoplasm of the GCs treated with FSH for 12 h. This indicated that the nuclear FOXO1 was re-localized to the cytoplasm of the cells upon the induction of FSH signaling. Conversely, this FSH-induced nuclear translocation of FOXO1 in the cultured GCs was mostly abolished by Ly294002 (Figure 4a). These findings verified that FSH promotes FOXO1 nuclear exclusion mediated by the PI3K/Akt signaling pathway in cultured GCs, which may result in changes in the effects of FOXO1 on ovarian follicle development and selection.

To further probe the biological roles of the FSH-induced phosphorylation and nuclear exclusion of FOXO1 in ovarian follicle development and growth, the expression of several key candidate genes, including *BCL2*, *CASP3*, *PCNA*, *CCND1,* and *FSHR,* that are tightly associated with GC proliferation, apoptosis, and follicle selection was determined. It has now been proven that the mRNA expression levels of the *BCL2*, *PCNA,* and *CCND1* genes that contribute to follicle development and growth by promoting cell proliferation and anti-apoptosis (*p* < 0.01 or *p* < 0.05; Figure 5b,d,e), as well as the *FSHR* mRNA that serves as the key biomarker for follicle selection, were significantly enhanced in the GCs under treatment with FSH (*p* < 0.01; as shown in the Figure 1b), while the expression level of *CASP3* mRNA was dramatically decreased (*p* < 0.05; Figure 5c). However, all of these stimulatory or inhibitory effects induced by FSH on the expression of these genes were effectively abolished by Ly294002 (*p* < 0.01; Figure 5b–e). These results indicated that FOXO1, a critical factor in promoting cell apoptosis, oppositely plays an indispensable role in animating GC proliferation in response to FSH-induced phosphorylation and nuclear exclusion during ovarian follicle growth and selection.

Based upon the molecular roles of the FSH-induced phosphorylation and nuclear exclusion of FOXO1 in the regulation of the aforementioned genes, flow cytometric analysis was used to explore the proliferation and apoptotic rate of cultured GCs with or without FSH treatment. As shown in Figure 4, a significantly higher cell proliferation rate and lower apoptotic rate of the GCs were found in the FSH treatment group compared to the control group (*p* < 0.05). Nevertheless, these FSH-induced effects on proliferation and apoptosis of GCs were significantly reduced by Ly294002 (*p* < 0.05, Figure 5f–k). These results validate the claim that the transcriptional factor FOXO1 has a critical role in governing GC proliferation or apoptosis via the FSH signaling-induced phosphorylation and nuclear exclusion of FOXO1 in ovarian follicle growth and selection of hens.

### 3.5. Crosstalk Between the PI3K/Akt and P62/Keap1/Nrf2 Signaling Pathways in Ovarian Follicle Growth and Selection Mediated by FOXO1 Factor

To further investigate the molecular mechanisms underlying FOXO1 nuclear exclusion in ovarian follicle growth and selection, the action of the P62/Keap1/Nrf2 signaling pathway in GC proliferation and the apoptosis of follicles was detected. As shown in Figure 6, the mRNA expression levels of *p62* and *Nrf2* were significantly diminished, while the expression level of *keap1* was markedly enhanced in the cells after 24 h of transfection with the pcDNA3,1(+)-FOXO1 vector (*p*  <  0.01), as determined by RT-PCR (Figure 6a–c). Furthermore, as the *p62* expression was efficiently knocked down in cells transfected with a p62-specific siRNA, as shown by RT-PCR, the mRNA expression level of *keap1* was noticeably fostered (*p* < 0.01, Figure 6b), while the expression level of *Nrf2* was markedly decreased in the cells (*p* < 0.01, Figure 6c). And a remarkable decrease in *p62* and *Nrf2* mRNA but a noted increase in *keap1* mRNA expression were observed (*p* < 0.01, Figure 6a–c). The results show that the overexpressed FOXO1 inhibits the expression of *p62* and *Nrf2* mRNA but promotes the expression of *keap1* mRNA, which indicated that the PI3K/Akt pathway may interact with the P62/Keap1/Nrf2 signaling pathway to play a critical role in response to FSH inducement in GC proliferation and follicle selection mediated by the FOXO1 factor.

To confirm this speculation, 740-Y-P was used to boost the level of FOXO1 phosphorylation and its sub-location in the cytoplasm of the GCs, as shown in Figure 3a–c, and LMB was added to the cultured GCs to attenuate FOXO1 phosphorylation and nuclear exclusion. Accordingly, the expression level of *p62* mRNA was notably increased in the cells under the administration of 740-Y-P (*p* < 0.01). Nevertheless, this effect was largely prevented by LMB (Figure 6d). Similarly, a dramatic increase in *BCL-2*, *PCNA,* and *CCND1* and a remarkable decrease in *CASP3* mRNA expression were observed (*p* < 0.01, Figure 6e–h). However, all these effects of 740-Y-P in the cells were abrogated by knockdown of *p62* expression. Moreover, it was found that the proliferation rate of the GCs with the 740-Y-P treatment was enhanced significantly (*p* < 0.01, Figure 6i–l), and the rate of apoptotic cells was markedly reduced (*p* < 0.01, Figure 6i–k,m), whereas all these outcomes of 740-Y-P treatment in GC proliferation or apoptosis were abolished by knockdown of *p62* expression (Figure 6l,m). The current data demonstrate that the FOXO1 factor plays a pivotal role in ovarian follicle growth and selection by mediating crosstalk between the PI3K/Akt and P62/Keap1/Nrf2 signaling pathways.

## 4. Discussion

In hen ovary, follicle selection (cyclic recruitment) consists of a single dominant follicle usually being selected into the preovulatory hierarchy on an approximate daily basis from a small cohort of prehierarchal follicles (small yellow follicles) sized 6–8 mm in diameter, followed by rapid growth and final differentiation; this is a bit different from other monovular species [5,48,49]. At this stage, the differentiating granulosa cells (GCs) from the selected follicles are characterized dramatically by the highest expression levels of *FSHR* mRNA, besides the structural and steroidogenic changes in these cells that cause the follicles to enlarge [11,15,50]. In fact, the highest levels of *FSHR* mRNA in the follicles were accordant with the functions of FSHR in maintaining the viability of prehierarchical follicles and in initiating GC differentiation immediately after the follicles are selected into the preovulatory hierarchy [1,50]. In response to ovarian FSH stimulation, previous studies have shown that FSH stimulates the production of more FSH receptors on the granulosa cells [51,52]. This allows the follicle to become more sensitive to FSH and grow more rapidly [53]. For instance, FSH treatment in vitro enhances the expression levels of *FSHR* mRNA in cultured GCs from human, rat, bovine, and chicken [51,54,55]. Moreover, treatment of granulosa cells for 48 h with FSH (1–100 ng/mL) increased *FSHR* mRNA levels in a dose-dependent manner in rat [52]. Similarly, in this study, our results have shown that FSH promotes the expression of *FSHR* mRNA in the GCs from hen ovarian prehierarchal follicles 6–8 mm in diameter in vitro and boosts follicle growth by stimulating GC proliferation and increasing resistance to apoptosis (Figure 1 and Figure 4). However, it should be noted that the effect of FSH on inducing FSHR expression is not universal and may be modulated by factors such as IGF-1, activin, or TGF-β signaling pathways [11,56,57]. The discrepancies among studies could be attributed to breed-specific differences in FSH responsiveness [58], distinct follicular developmental stages [59], variations in FSHR genotypes [60], and the presence of local ovarian factors such as IGF-1 or cAMP [61,62,63,64,65,66].

It is generally accepted that FSH is a key regulatory factor in follicular development, with its actions being mediated and finely tuned by intraovarian factors [11]. FSH/FSHR signaling can activate multiple downstream pathways, including cAMP/PKA and PI3K/Akt, which collectively regulate granulosa cell proliferation, differentiation, and steroidogenesis [67,68]. Previous studies have demonstrated that the PI3K/Akt signaling pathway plays a pivotal role in granulosa cell proliferation and apoptosis [56,59]. The PI3K/Akt-mediated phosphorylation/inhibition of FOXO1 is required to relieve FOXO1′s inhibitory influence on GC proliferation and differentiation [13,28,60,63]. Furthermore, phosphorylation of FoxOs induced by the FSH-PI3K/Akt signaling pathway results in the exclusion of the FoxO factors from the nucleus to the cytosol and inhibits FoxO-dependent transcription [26,27]. In this study, we found that FSH-induced phosphorylation of FOXO1 in hen ovarian GCs may be mainly attributed to the two sites, Ser^248^ and Ser^311^, of the three deduced residues (Figure 1). Meanwhile, the phosphorylation of FOXO1 at the sites may be closely associated with an enhanced nuclear exclusion of the p-FOXO1 protein. However, it was unexpectedly found that there is a difference between the number of the phosphorylated residues (Ser^248^ and Ser^311^) on chicken FOXO1 and that of the phosphorylated residues (Thr^24^, Ser^256^, and Ser^319^) on human FoxO1 [32], and the three amino acid residues (Thr^24^, Ser^253^, and Ser^316^) on mouse FoxO1 in the ovarian GCs [67,69], which are induced by FSH and mediated via the PI3K/Akt signaling pathway, respectively.

Previous studies in mammals have indicated that FoxO1 is involved in GC apoptosis induced by FSH via mediation of the PI3K/Akt pathway [13,27]. To explore the precise effects of the phosphorylated sites (Ser^248^ and Ser^311^) on the p-FOXO1 factor in chicken follicular GCs mediated by the FSHR/PI3K/Akt signaling pathway, FSH (10 ng/mL) was administered to the medium of the GC culture for 12 h with or without Ly294002 (25 μM), a PI3K inhibitor to prevent the PI3K/Akt pathway [70,71,72]. These results strongly supported that the FSH-induced phosphorylation of FOXO1 in ovarian GCs was predominantly determined by the two sites, Ser^248^ and Ser^311^, and mediated by the FSHR/PI3K/Akt signaling pathway. Additionally, the phosphorylation of FOXO1 may directly lead to its nuclear exclusion, which caused the reduction of FOXO1 expression by the enhanced degradation of the p-FOXO1 protein in the cytoplasm of the GCs. Therefore, the two phosphorylated sites on chicken FOXO1 may be the unique characteristics different from those of the FoxO1 with the three sites in mammals; this finding reveals an important mechanistic difference between chickens and mammals in molecular regulation. In addition, FSH can promote the acetylation of FOXO1 through the cAMP/PKA signaling pathway, thereby enhancing the phosphorylation of FOXO1 at Ser248 and suggesting crosstalk between the PKA and PI3K/Akt pathways. Such integration of multiple signaling cascades may ensure precise control of granulosa cell fate during follicle selection.

To understand the regulatory mechanisms underlying the nuclear exclusion of FOXO1 induced by PI3K/Akt-dependent phosphorylation at these sites, co-immunoprecipitation (Co-IP) analysis was performed to determine which proteins were involved in these actions, such as Ran, CRM1, and 14-3-3 proteins, which have been reported to play a critical role in nucleocytoplasmic transport in mammals [34,36,73]. Among them, Ran is an evolutionarily conserved member of the Ras superfamily that regulates all receptor (e.g., nuclear exportin CRM1)-mediated transport between the nucleus and the cytoplasm [73]. CRM1 and RanGTP attach to the mouse FoxO protein via interactions with the NES (nuclear export sequence) [74]. The nuclear RanGTP promotes cargo-binding to exportins, including CRM1, which export cargo from the nucleus [75], which is consistent with our conclusion that the activated Ran enhanced the combination of FOXO1 and CRM1 in a PI3K/Akt signaling-independent manner, but the two PKB phosphorylation sites in FOXO1 are indispensable for their combination with Ran. Concerning the roles of 14-3-3 in the nuclear exclusion of FOXO1, previous studies have shown that the 14-3-3 proteins are a family of dimeric regulatory proteins that are involved in many biologically important processes by binding to other proteins in a phosphorylation-dependent manner [76,77]. Chicken FOXO1 has two PKB motifs (one at the C-terminal, the other in the forkhead domain) that interact with the 14-3-3 proteins [77], respectively. Phosphorylation at three highly conserved PKB phosphorylation sites of FoxO results in its sequestration in the cytosol by 14-3-3 proteins [34,35]. In this study, our results demonstrated that the combination of FOXO1 and 14-3-3 mainly depended on FOXO1 phosphorylation induced by the PI3K/Akt signaling pathway but not influenced by Ran activity, which results in FOXO1 extrusion from the nucleus of the GCs.

Beyond its role in promoting cell proliferation, the PI3K/Akt signaling pathway may also interact with stress-related signaling cascades. In the present study, our results revealed that FSH boosts the expression of *FSHR* mRNA and promotes FOXO1 nuclear exclusion regulated by the PI3K/Akt signaling pathway in cultured GCs, leading to an increase in GC proliferation and inhibition of apoptosis in the prehierarchical follicles of 6–8 mm in diameter of hen ovary, indicating that the FOXO1 factor serves as an effector of the PI3K/Akt signaling pathway, and the FSH-induced nuclear exclusion of FOXO1 modulated by the pathway in the GCs is closely correlated with follicle selection and growth. As previously mentioned, the p62/Keap1/Nrf2 pathway plays a critical role in the regulation of mammalian ovarian granulosa cell apoptosis and ferroptosis, as well as oxidative stress and excessive autophagy [37,38,39]. Although the canonical Keap1/Nrf2 signaling pathway is implicated in adult chicken ovarian functions, which is modulated by estrogen receptor alpha (ERα), and p62/SQSTM1, a selective autophagy adaptor protein, was reported to regulate GC differentiation and antral follicle formation via FSH induction in female mice [78], the potential effects and molecular mechanism of FOXO1 and p62 on GC proliferation and apoptosis of follicles of 6–8 mm in diameter of hen ovary by targeting the Keap1/Nrf2 signaling pathway remains ambiguous. Therefore, this study was conducted to explore the roles of the p62/Keap1/Nrf2 pathway in follicle selection and growth, and its interactions with the PI3K/Akt pathways. Accordingly, the results showed that overexpressed FOXO1 inhibits the expression of *p62* and *Nrf2* mRNA but promotes the expression of *keap1* mRNA. Moreover, the proliferation rate of GCs with 740-Y-P treatment was significantly enhanced, and the rate of apoptosis in the cells was markedly reduced, whereas all these outcomes of 740-Y-P treatment in GC proliferation or apoptosis were abolished by knockdown of *p62* expression, which indicated that the FOXO1 factor plays a pivotal role in ovarian follicle growth and selection by bridging crosstalk between the PI3K/Akt and P62/Keap1/Nrf2 signaling pathways in chickens. The findings of the present study contribute to a comprehensive understanding of the molecular mechanisms and biological roles of the FSH-induced nuclear exclusion of FOXO1 mediated by the PI3K/Akt signaling pathway in granulosa cells, which are involved in follicle selection and growth of the hen ovary (Figure 7). This finding provides new insights into the molecular basis of the FSH-mediated regulation of follicular development and identifies potential molecular targets for improving follicle selection efficiency and enhancing laying performance in chickens.

## 5. Conclusions

The present data demonstrated that the FSH-induced nuclear exclusion of FOXO1 is closely associated with hen ovarian follicle selection and growth by promoting higher *FSHR* mRNA expression and GC proliferation and inhibiting cell apoptosis via the PI3K/Akt signaling pathway in vitro. Moreover, we first proved that the PI3K/Akt-dependent phosphorylation of chicken FOXO1 mainly depends on the two sites, Ser^248^ and Ser^311^, and results in the nuclear exclusion of FOXO1, in which activated Ran, CRM1, and 14-3-3 are implicated as nucleocytoplasmic transport factors. In this process, FOXO1 serves as both a key intracellular signal transducer for PI3K/Akt signaling and a mediator to bridge crosstalk between the PI3K/Akt and P62/Keap1/Nrf2 signaling pathways.

## Figures and Tables

**Figure 1 cells-14-01864-f001:**
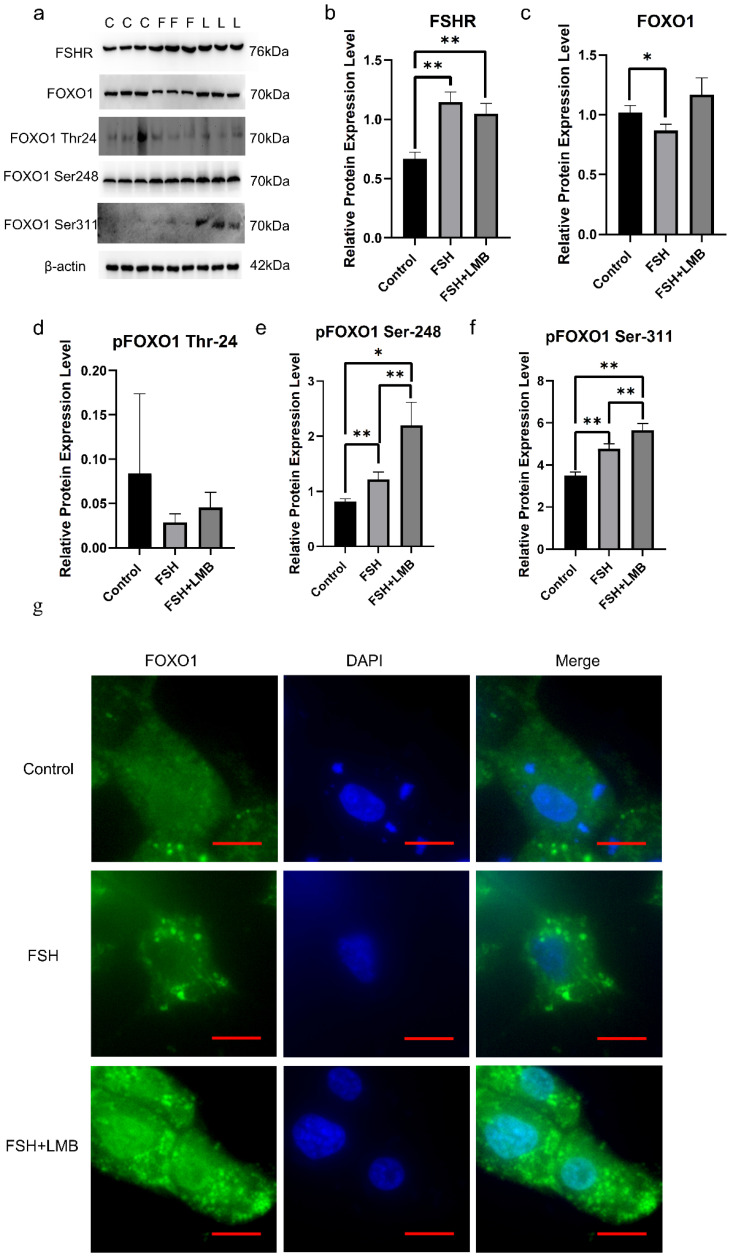
Effects of FSH on phosphorylation and nuclear exclusion of FOXO1 in ovarian GCs. (**a**) The expression of FSHR, FOXO1, and phosphorylated FOXO1 (p-FOXO1) corresponding to the sites Thr^24^, Ser^248^, and Ser^311^ in cultured GCs under treatment with FSH or/and LMB was determined via Western blotting by using the anti-FSHR, anti-FOXO1, and anti-p-FOXO1 corresponding to Thr^24^, Ser^248^, and Ser^311^, respectively. The β-actin was used as a loading control. All blots were cropped, and the gels were run under the same experimental conditions. (**b**) The expression levels of FSHR protein in cultured GCs under FSH or/and LMB treatment by Western blotting, as shown in (**a**). (**c**) Expression levels of FOXO1 under treatment the same as (**b**). (**d**) The expression levels of p-FOXO1 corresponding to the site Thr^24^. (**e**) The expression levels of p-FOXO1 corresponding to the site Thr^248^. (**f**) The expression levels of p-FOXO1 corresponding to the site Thr^311^. For each group, the superscript symbol above the bar indicates that the difference was significant compared to the control group, ** *p* < 0.01, * *p* < 0.05. (**g**) The subcellular localizations of FOXO1 protein in cultured GCs under FSH or/and LMB treatment by immunofluorescence staining method. The red line segment at the bottom right corner of the image is the scale bar (Leica, 400×; scale bar = 5 µm).

**Figure 2 cells-14-01864-f002:**
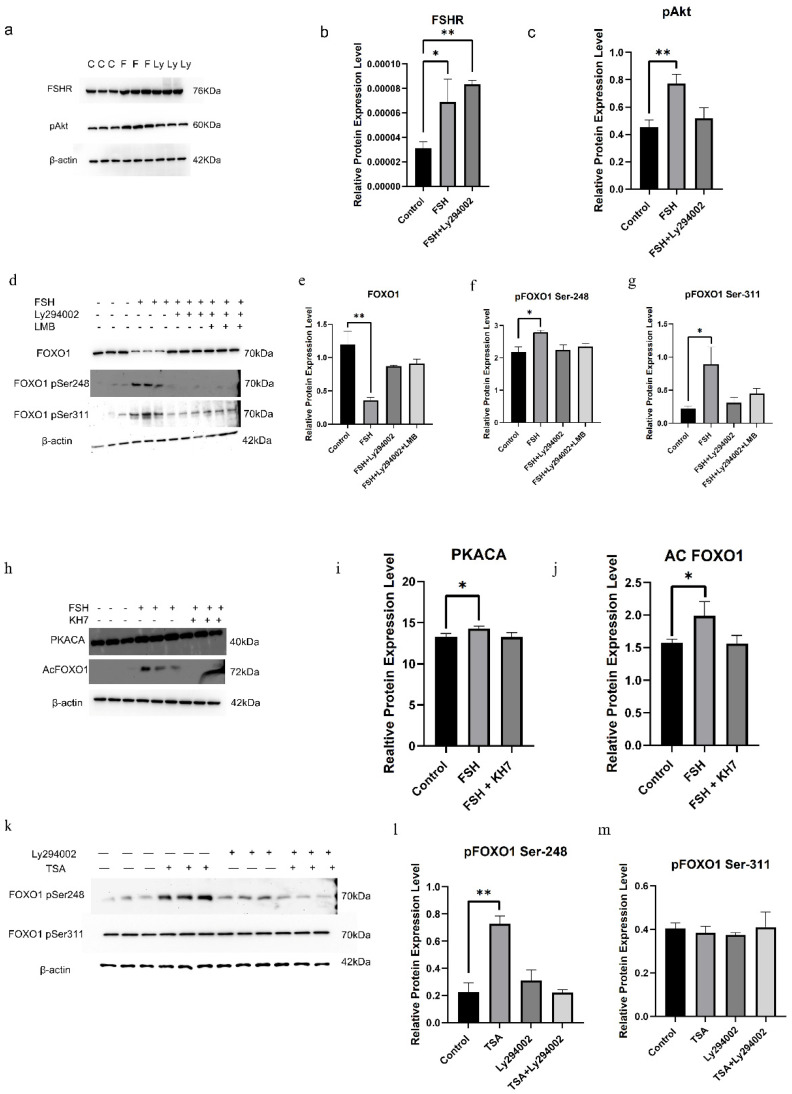
The roles of activated PI3K/Akt signaling in the FSH-induced phosphorylation of FOXO1 in GCs. (**a**) The expression of FSHR and phosphorylated Akt (p-Akt) in cultured GCs under treatment with FSH or/and Ly294002 was determined by Western blotting. (**b**) The expression levels of FSHR protein in GCs under FSH or/and Ly294002 treatment are shown in (**a**). (**c**) The expression levels of p-Akt protein in GCs under the same treatment as (**b**). (**d**) The expression of FOXO1, p-FOXO1 corresponding to the phosphorylation site, Se^r248^ or Ser^311^, in cultured GCs under treatment with FSH or/and Ly294002 and LMB was examined by Western blotting, respectively. (**e**) The expression levels of FOXO1 protein in cultured GCs. (**f**) The expression levels of pFOXO1 corresponding to the Ser^248^ site. (**g**) The expression levels of pFOXO1 corresponding to the Ser^311^ site. (**h**–**j**) The expression levels of PKACA and acetylated FOXO1 (Ac-FOXO1) in cultured GCs under treatment with FSH or/and KH7 were tested by Western blotting, respectively. (**k**–**m**) The expression levels of the pFOXO1 proteins corresponding to the site, Ser^248^ or Ser^311^, in cultured GCs under treatment with Ly294002 or/and TSA were determined by Western blotting, respectively. β-actin was used as a loading control. All blots were cropped, and the gels were run under the same experimental conditions. For each group, the superscript symbol above the bar indicates that the difference was significant compared to the control group, ** *p* < 0.01, * *p* < 0.05.

**Figure 3 cells-14-01864-f003:**
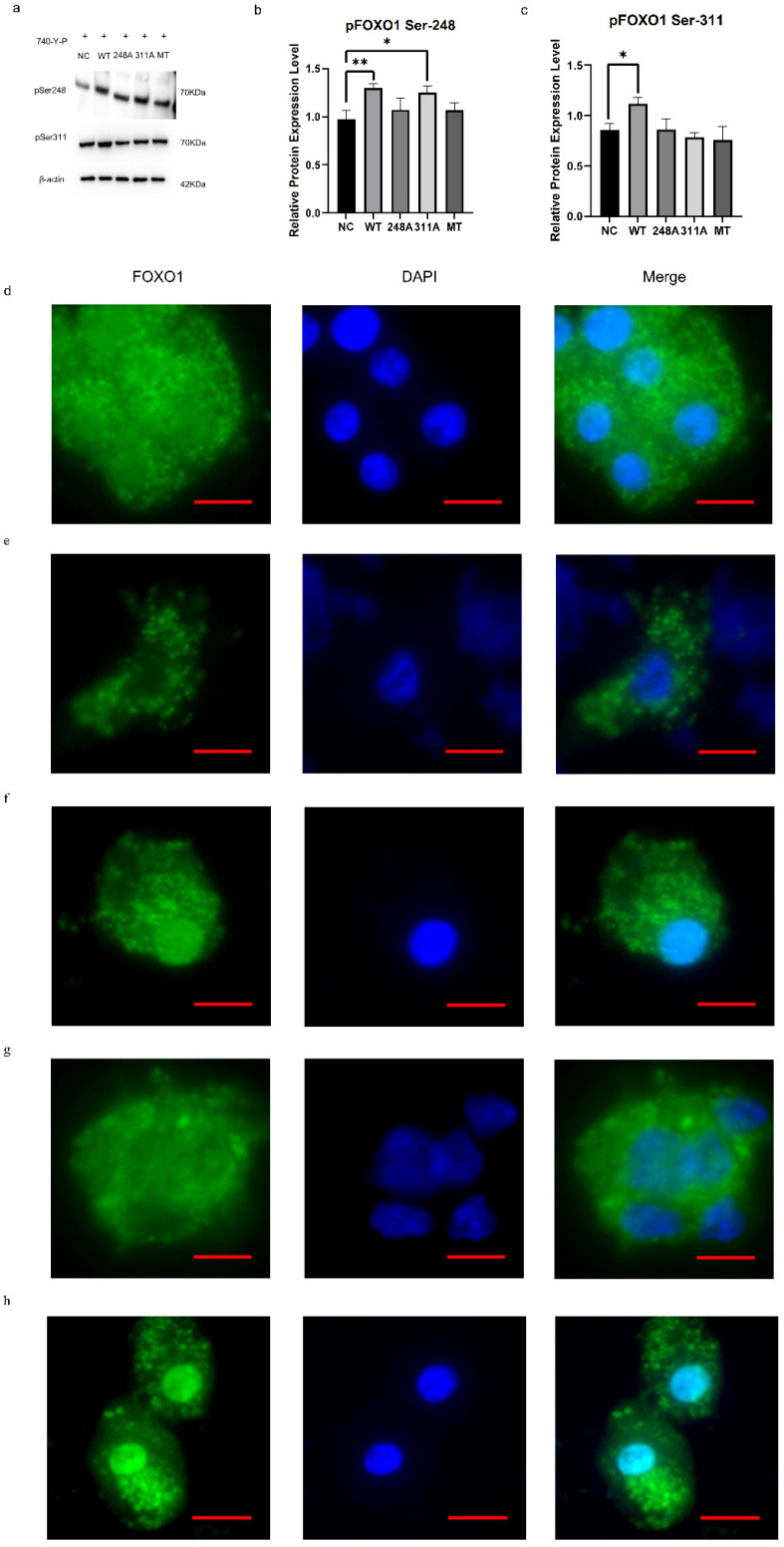
Roles of the FOXO1 phosphorylation sites (Ser^248^ and Ser^311^) in regulating its nuclear exclusion. (**a**) The overexpression of the p-FOXO1 corresponding to the phosphorylation sites, Ser^248^ and Ser^311^ wild-type or mutation-type (S248A or/and S311A), in cultured GCs under treatment with 740-Y-P was determined by Western blotting, respectively. NC denotes negative control group. WT indicates the overexpression group of FOXO1 wild-type vector; 248A represents the group of overexpressed S248A FOXO1 mutant vector; and 311A denotes the overexpressed S311A FOXO1 mutant group. MT indicates the overexpression vector of FOXO1 with the S248A and S311A mutation. β-actin was used as a loading control. All blots were cropped, and the gels were run under the same experimental conditions. (**b**) The expression levels of p-FOXO1 corresponding to the Ser^248^ site. (**c**) The expression levels of pFOXO1 corresponding to the Ser^311^ site in the cultured GCs. (**d**) The subcellular localizations of FOXO1 observed in the cultured GCs by immunofluorescence staining in the NC group. (**e**) The subcellular localizations of FOXO1 in the WT. (**f**) The subcellular localizations of FOXO1 in the 248A group. (**g**) The subcellular localizations of FOXO1 in the 311A group. (**h**) The subcellular localizations of FOXO1 in the MT group. The red line segment at the bottom right corner of the image is the scale bar (Leica, 400×; scale bar = 5 µm). ** *p* < 0.01, * *p* < 0.05.

**Figure 4 cells-14-01864-f004:**
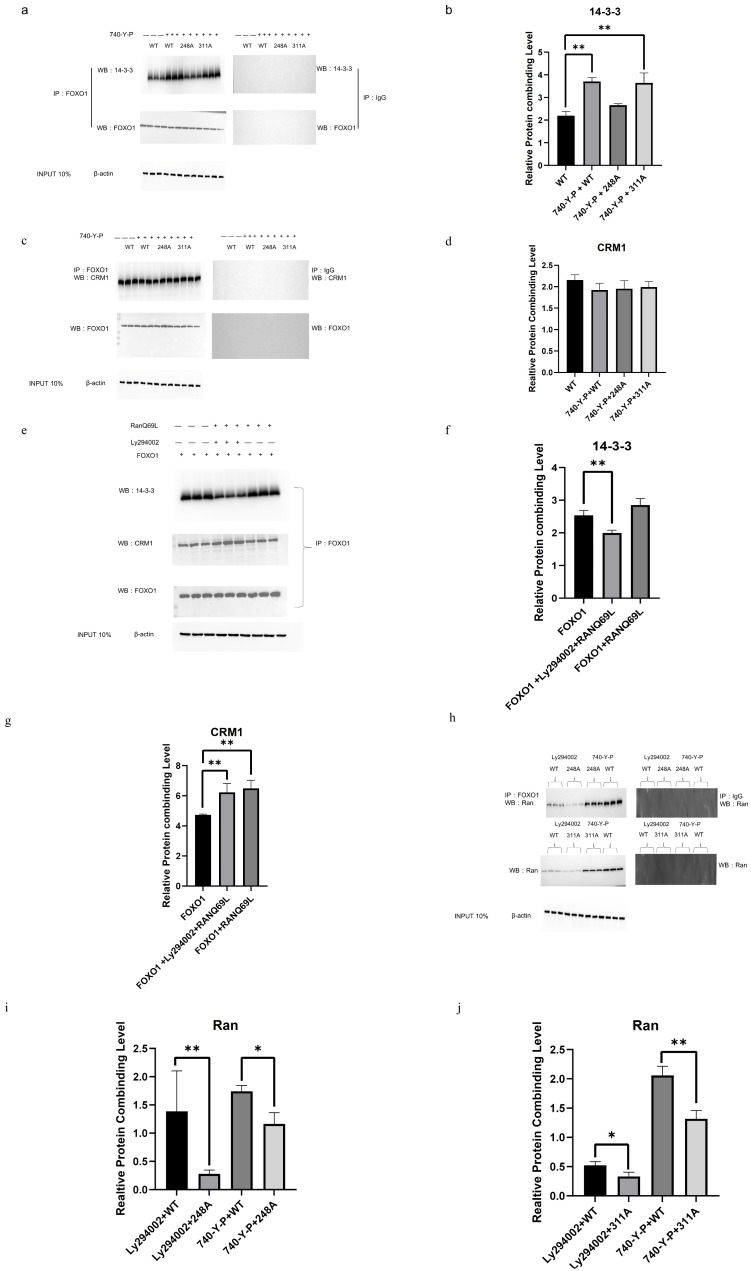
Involvement of the Ran, CRM1, and 14-3-3 proteins in the nuclear exclusion of FOXO1. (**a**) Effects of FOXO1 mutant with the S248A or/and S311A substitution on its combination with 14-3-3 determined by co-immunoprecipitation in presence or absence of 740-Y-P. The chicken GCs were transfected with the expression constructs of the pcDNA3,1(+)-FOXO1 wild-type, pcDNA3,1(+)-FOXO1 Ser248A mutant, and pcDNA3,1(+)-FOXO1 Ser311A mutant, respectively. (**b**) The protein combining level of 14-3-3 with FOXO1 protein in the cultured GCs. (**c**) Effects of FOXO1 mutant with the S248A or/and S311A substitution on its combination with the CRM1 determined by co-immunoprecipitation in presence or absence of 740-Y-P. (**d**) The combining level of CRM1 with FOXO1 protein in the cells. (**e**) Effects of RANQ69L overexpression on the combination of FOXO1 with 14-3-3 or CRM1 examined by co-immunoprecipitation in presence or absence of Ly294002. The chicken GCs were transfected with the pcDNA3,1(+)-FOXO1 expression construct, and simultaneously with the pcDNA3,1(+)- RANQ69L plasmid. (**f**) The protein combining level of 14-3-3 with FOXO1 in the cells under FOXO1 overexpression with or without Ly294002 treatment and with or without RANQ69L treatment by using Western blotting. (**g**) The protein combining level of CRM1 with FOXO1 under the same conditions as (**f**). (**h**). Effects of the two PKB phosphorylation sites in FOXO1 on its combination with RAN. The ovarian GCs were transfected with the expression constructs, pcDNA3,1(+)-FOXO1 wild-type, pcDNA3,1(+)-FOXO1 Ser248A mutant, and pcDNA3,1(+)-FOXO1 mutant, in presence or absence of 740-Y-P or in presence or absence of Ly294002, respectively, which were determined by the co-immunoprecipitation assay. (**i**) The combination levels of Ran with FOXO1 in cultured GCs transfected by the plasmid of pcDNA3,1(+)-FOXO1 Ser248A mutant with/without 740-Y-P treatment or with/without Ly294002 treatment. (**j**) The combination levels of Ran with FOXO1 in the cultured GCs transfected by the plasmid of pcDNA3,1(+)-FOXO1 Ser311A mutant with/without 740-Y-P treatment or with/without Ly294002 treatment. *n* = 3. ** *p*  <  0.01, * *p*  <  0.05.

**Figure 5 cells-14-01864-f005:**
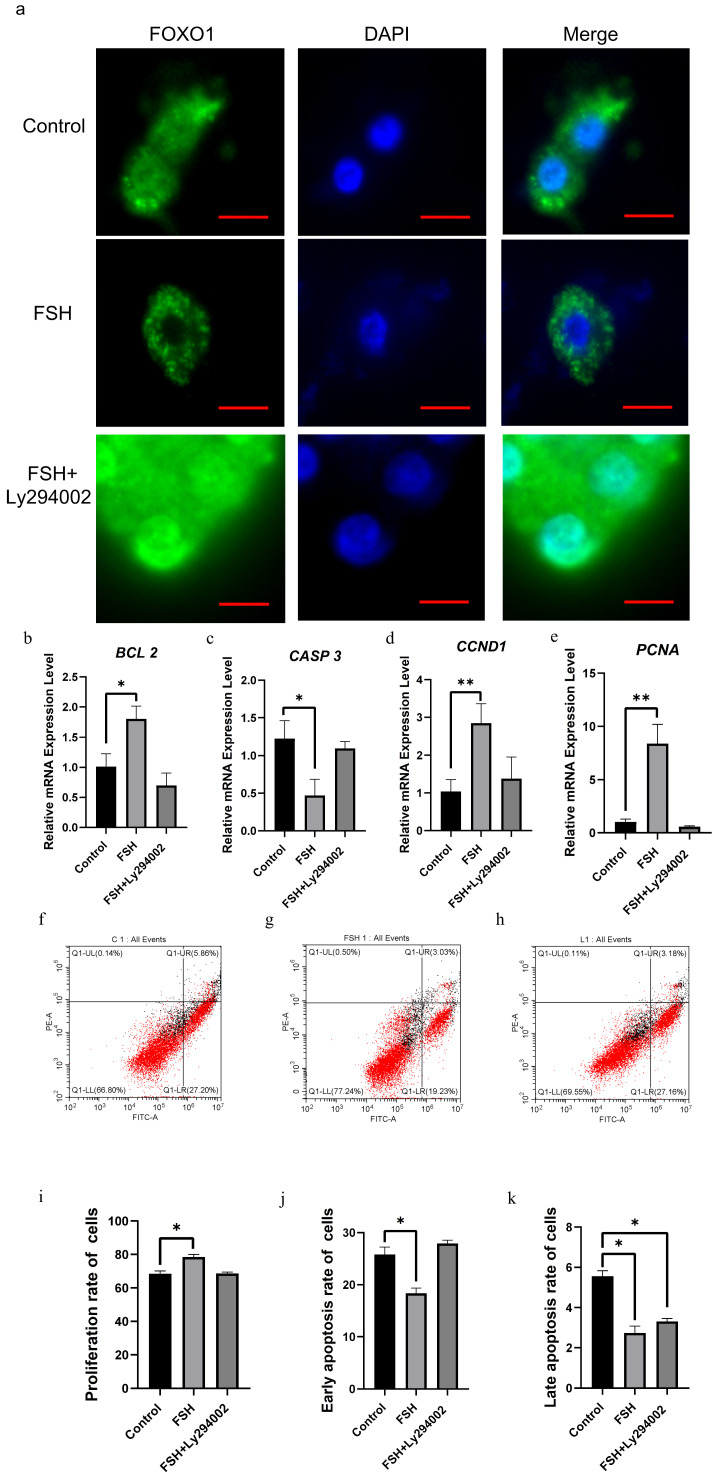
Effects of FSH-induced FOXO1 nuclear exclusion on GC proliferation and apoptosis via PI3K/Akt signaling pathway. (**a**) The subcellular localizations of FOXO1 protein in cultured GCs under FSH or/and Ly294002 treatment by immunofluorescence assay. The red line segment at the bottom right corner of the image is the scale bar (Leica, 400×; scale bar = 5 µm). (**b**) The expression levels of *BCL* mRNA in cells under FSH or/and LMB treatment by RT-qPCR assay. (**c**) The expression levels of *CASP3* mRNA under the same treatment as (**b**). (**d**) The expression levels of *CCND1* mRNA. (**e**) The expression levels of *PCNA* mRNA. (**f**–**k**) The GC proliferation and apoptosis under FSH or/and LMB treatment by flow cytometry assay. All data are presented as means ± SEM. *n* = 3. For each group, the superscript symbol above the bar indicates that the difference was significant compared to the control group, ** *p*  <  0.01, * *p*  <  0.05.

**Figure 6 cells-14-01864-f006:**
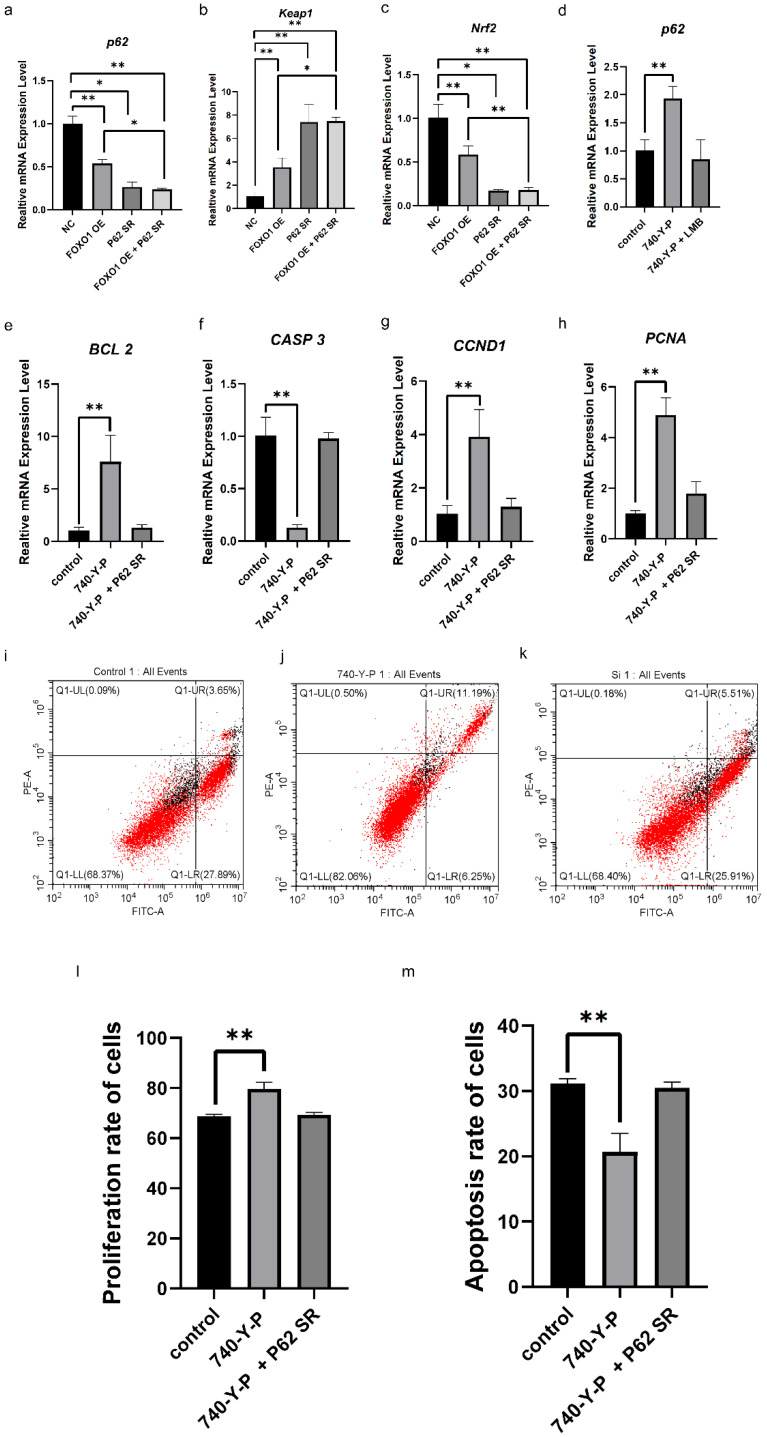
Crosstalk of PI3K/Akt and P62/Keap1/Nrf2 pathways in regulating GC proliferation and apoptosis mediated by FOXO1. (**a**) Expression levels of *P62* mRNA under FOXO1 overexpression or/and *P62* knockdown examined by RT-qPCR assay. NC: negative control, OE: overexpression, SR: siRNA. The mRNA expression was normalized to that of the *18S* rRNA gene; the values of the bar graphs represent the mean  ±  SEM of 3 hens (*n*  = 3) from a representative experiment. (**b**) The expression levels of *Keap1* mRNA under the same condition as (**a**). (**c**) The expression levels of *Nrf2* mRNA under the same conditions as (**a**). (**d**) The expression levels of *p62* mRNA under 740-Y-P or/and LMB treatment by RT-qPCR assay. (**e**) The expression levels of *BCL2* mRNA under 740-Y-P treatment or/and P62 knockdown by RT-qPCR. (**f**–**h**) The expression level results of *CASP3*, *CCND1,* and *PCNA* mRNA under the same conditions as (**e**). (**i**–**m**). GC proliferation and apoptosis under treatment with FSH or/and P62 knockdown by flow cytometry assay. All data are presented as the means ± SEM. *n* = 3. For each group, the superscript symbol above the bar indicates that the difference was significant compared to the control group, ** *p*  <  0.01, * *p*  <  0.05.

**Figure 7 cells-14-01864-f007:**
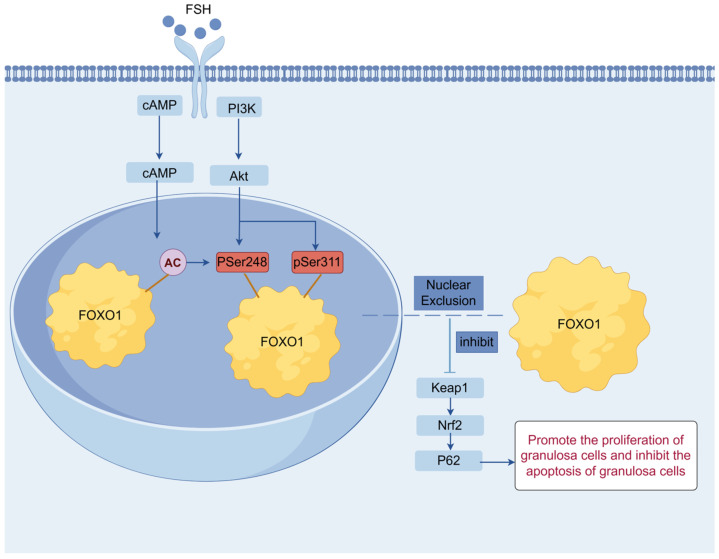
Schema illustrating the mechanism of regulation of chicken ovarian functions by FSH-induced nuclear exclusion of FOXO1 via the PI3K/Akt signaling pathway in granulosa cells.

**Table 1 cells-14-01864-t001:** Antibodies used for Western blotting assay.

Antibody	Dilution	Manufacturer	Item Number
anti-β-actin antibody	1:1000	Bosterbio, Wuhan, China	MO1263-4
rabbit anti-phospho-FOXO3a (Ser319) antibody	1:1000	Affinity Biosciences, Cincinnati, OH, USA	AF3418
rabbit anti-phospho-FOXO4 (Thr24) antibody	1:1000	Affinity Biosciences, Cincinnati, OH, USA	bs-3145R
rabbit anti-phospho-FOXO4 (Ser256) antibody	1:1000	ZenBio Science, Cupertino, CA, USA	310198
rabbit anti-FOXO1 antibody	1:1000	Proteintech Group, Chicago, IL, USA	18592-1-AP
rabbit anti-FSHR antibody	1:1000	Proteintech Group, Chicago, IL, USA	10849-1-AP
rabbit anti-acetyl-FOXO1 antibody	1:1000	Immunoway, Suzhou, China	YK0110
HRP-conjugated affinipure goat anti-Mouse	1:2000	Proteintech Group, Chicago, IL, USA	SA00001-1
HRP-conjugated affinipure Goat anti-rabbit	1:2000	Proteintech Group, Chicago, IL, USA	SA00001-2
rabbit anti-pAkt antibody	1:1000	Proteintech Group, Chicago, IL, USA	P31749
rabbit anti-PKACA antibody	1:1000	Bioss Antibodies, Beijing, China	bs-0520R
rabbit anti-CRM1 antibody	1:1000	Bioss Antibodies, Beijing, China	bs-3145R
rabbit anti-14-3-3 antibody	1:1000	Proteintech Group, Chicago, IL, USA	10849-1-AP
rabbit anti-RAN antibody	1:1000	Proteintech Group, Chicago, IL, USA	10469-1-AP

**Table 2 cells-14-01864-t002:** Primer pairs designed for quantitative real-time PCR analysis.

Name	Primer	Primer Sequences (5′-3′)	Accession No.	Product Size
*proFOXO1*	Forward	ACCCATCATCAGCCACCAAA	NM_204328.2	146 bp
	Reverse	AGCAGATGACGACTGGGTTG		
*18S* rRNA	Forward	TAGTTGGTCGAGCGATTTGTCT	AF173612.1	169 bp
	Reverse	CGGACATCTAAGGGCATCACA		
*Bcl2*	Forward	CCGCTACCAGAGGGACTT	NM_205339.3	155 bp
	Reverse	ACATCACGCCGCCGAAC		
*Caspase3*	Forward	AAGAACTTCCACCGAGATACCG	XR_006936397.1	204 bp
	Reverse	GCTTAGCAACACACAAACAAAA		
*CCND1*	Forward	ATAGTCGCCACTTGGATGCT	NM_205381.2	230 bp
	Reverse	TCGGGTCTGATGGAGTTGTC		
*PCNA*	Forward	CTGAGGCGTGCTGGG	NM_204170.3	133 bp
	Reverse	ATGGCGATGTTGCGG		
*P62*	Forward	CCAGGAACACAGCGAGTCAA		152 bp
	Reverse	CACCCTCATCAGAGAAGCCC		

## Data Availability

All data used and analyzed during the present study are included in this published article and its Appendix A.

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
