# Peer review of "FSH-Induced Nuclear Exclusion of FOXO1 Mediated by PI3K/Akt Signaling Pathway in Granulosa Cells Is Associated with Follicle Selection and Growth of the Hen Ovary"

_cells, 2025, doi:10.3390/cells14231864_

Round 1
Reviewer 1 Report
Comments and Suggestions for Authors
The subject of the study is interesting, the methodology is adequate and the obtained results are numerous. Nevertheless, the current version of manuscript has serious formal issues, which does not enable me to recommend their publication in its present form:
ABSTRACT should contain the clearly formulated hypothesis, novelty of the study, experimental protocol, analytical methods and separate Conclusions. It is not a case. l.19-29. in birds or mammals?
INTRODUCTION. By description of the current state of art please indicate species and method which was used as a basis of each statement. It is especially important to enable to understand the novelty of the study (whether the present results are novel for chicken, birds, mammals or for any species). As in Abstract, the hypothesis which the authors try to validate and the novelty of the performed study should be clearly formulated. The reasons, why just this model has been used and just these parameters were measured should be indicated. Demonstration of novelty of the present data and their differences with data published previously is especially important because similar results have been already submitted by these authors to another journal.
MATERIALS AND METHODS. Why not the preovulatory, but 8-8 mm follicles were used? What was age of hens? Each publication should enable the reader to understand and to repeat the described study. The manuscript does not enable it. The manufacturers (name, city, country, cat.no) of each facility and consumable used should be indicated. Positive and negative control for each method too. Statistics: the number of experiments, number of analyzed samples per group in each experiment is not indicated. Student’s t-test might be used only if the normal distribution of values has been demonstrated. It is not a case too.
RESULTS. Elements of Discussion (interpretation of the results) should be excluded or minimalized.
DISCUSSION should be structuralized according to the query to be addressed by performed study. Discussion of the particular problem/query should (1) briefly summarize the results obtained by the authors, (2) compare they with the data obtained previously, (3) to stress the novelty, contribution of the data obtained by author’s study to theory and praxis, (4) to outline the hypothesis concerning biological mechanisms of process integrating all the available (author’s and previous) data, (5) to formulate possible hypothesis and suggestions concerning practical application of this hypothesis and (6) to state the weak points of the performed study and to outline the direction of the future studies. It is not a case. The schema illustrating the mechanism of regulation of chicken ovarian functions by FSH and FOXOs based on the results of the author’s study would be very helpful. Significance of the obtained results for general biology of reproduction and praxis (poultry breeding) and their novelty should be shown.
GENERAL. I cannot understand, why some words in the text are marker in red.
Author Response
|
1. ABSTRACT should contain the clearly formulated hypothesis, novelty of the study, experimental protocol, analytical methods and separate Conclusions. It is not a case. l.19-29. in birds or mammals? Response: Thank you for your suggestions. I have added the experimental design and research conclusions to the abstract of the manuscript.The characteristics described in Lines 12-19 are those of signaling pathways in mammals, which is mentioned in Line 21.
2. INTRODUCTION. By description of the current state of art please indicate species and method which was used as a basis of each statement. It is especially important to enable to understand the novelty of the study (whether the present results are novel for chicken, birds, mammals or for any species). As in Abstract, the hypothesis which the authors try to validate and the novelty of the performed study should be clearly formulated. The reasons, why just this model has been used and just these parameters were measured should be indicated. Demonstration of novelty of the present data and their differences with data published previously is especially important because similar results have been already submitted by these authors to another journal. Response: Thank you very much. We have revised the Introduction according your suggestions in the manuscript.
3. MATERIALS AND METHODS. Why not the preovulatory, but 8-8 mm follicles were used? What was age of hens? Each publication should enable the reader to understand and to repeat the described study. The manuscript does not enable it. The manufacturers (name, city, country, cat.no) of each facility and consumable used should be indicated. Positive and negative control for each method too. Statistics: the number of experiments, number of analyzed samples per group in each experiment is not indicated. Student’s t-test might be used only if the normal distribution of values has been demonstrated. It is not a case too. Response: Thank the reviewer for these important comments. ①The diameter range of 6-8 mm represents a critical stage for prehierarchical follicles to develop into hierarchical follicles through follicle selection. So we chose follicles with a diameter of 6 to 8 mm for the experiment. ②The age of the chicken is 21 weeks, we have added it in the MATERIALS AND METHODS section. ③The Materials and Methods section has been thoroughly revised to specify the manufacturers (name, city, country) and catalog numbers for all reagents, instruments, and consumables used in the experiments. ④Positive and negative control for each method too. In this study, all experiments used either the negative control group treated with PBS or the group transfected with an empty vector as the control group, and the specific treatment methods have been described in Sections 2.2 and 2.3. ⑤And all experiments in this article included 3 replicates per group. We have re-examined the data distribution. The normality of each dataset was tested using the Shapiro–Wilk test before applying Student’s t-test. Only data showing normal distribution were analyzed with the t-test.
4. RESULTS. Elements of Discussion (interpretation of the results) should be excluded or minimalized. DISCUSSION should be structuralized according to the query to be addressed by performed study. Discussion of the particular problem/query should (1) briefly summarize the results obtained by the authors, (2) compare they with the data obtained previously, (3) to stress the novelty, contribution of the data obtained by author’s study to theory and praxis, (4) to outline the hypothesis concerning biological mechanisms of process integrating all the available (author’s and previous) data, (5) to formulate possible hypothesis and suggestions concerning practical application of this hypothesis and (6) to state the weak points of the performed study and to outline the direction of the future studies. It is not a case. The schema illustrating the mechanism of regulation of chicken ovarian functions by FSH and FOXOs based on the results of the author’s study would be very helpful. Significance of the obtained results for general biology of reproduction and praxis (poultry breeding) and their novelty should be shown. Response: Thank you very much. We have revised it according your suggestions in the manuscript. And the schema illustrating the mechanism of regulation of chicken ovarian functions by FSH and FOXOs has been added in Figure 7. 5. GENERAL. I cannot understand, why some words in the text are marker in red. Response: The red-colored text in the manuscript was used to mark key points during writing process. We sincerely apologize for failing to correct this before the formal submission, and it has been corrected.
|
Reviewer 2 Report
Comments and Suggestions for Authors
The study explores the molecular mechanisms by which FSH regulates FOXO1 phosphorylation and nuclear exclusion in granulosa cells of the hen ovary through the PI3K/Akt pathway, and how this contributes to follicle selection and growth. The topic is relevant to reproductive biology and avian physiology, and the experiments are extensive, including protein expression analyses, site-directed mutagenesis, co-immunoprecipitation, and functional assays for cell proliferation and apoptosis. The research provides novel mechanistic insight into FSH–PI3K/Akt–FOXO1 signaling in the avian ovary, an area less studied than mammalian systems. The experiments are logically connected and well-supported by quantitative and imaging data. The link between FOXO1 and the P62/Keap1/Nrf2 pathway adds an interesting dimension connecting hormonal signaling with redox regulation.
Major points to improve:
English and Style: the manuscript requires substantial language polishing to improve clarity, conciseness, and readability. Some sentences are overly long, repetitive, or contain grammatical inconsistencies. Professional English editing is strongly recommended.
Introduction: while comprehensive, it should be more focused. Several paragraphs repeat general knowledge and contain excessive citations; the rationale and specific research hypothesis could be stated more clearly.
Methods: although detailed, the section would benefit from better organization (e.g., concise subsections, removal of redundant manufacturer details). The statistical approach should specify n, type of post hoc tests, and confidence intervals.
Results presentation: figures are numerous and informative, but labeling (e.g., “a–h”) and legends could be simplified for readability. Including a schematic summary of the proposed mechanism would help readers.
Discussion: this section is long and partially repetitive of the Results. Please focus on novel findings, limit citations to essential studies, and discuss the potential physiological implications for follicle hierarchy in hens.
Mechanistic claims: while the experiments convincingly show PI3K/Akt-mediated phosphorylation of FOXO1, the suggested crosstalk with the p62/Keap1/Nrf2 pathway remains correlative and requires cautious interpretation. The term “completely mediated” or “directly results in” should be softened unless supported by additional functional assays (e.g., pathway inhibition combined with rescue).
Minor comments:
Gene and protein symbols should be italicized or capitalized according to standard nomenclature.
Figures should include scale bars and molecular weight markers for Western blots.
The abbreviations list is missing or incomplete.
Some references are outdated or duplicated; please check citation accuracy and formatting according to Cells guidelines.
Comments on the Quality of English LanguageThe English language in the manuscript requires substantial editing. While the text is understandable, it contains overly long and repetitive sentences, grammatical inconsistencies, and awkward phrasing; professional language polishing is recommended to improve clarity, conciseness, and scientific precision.
Author Response
|
1. Introduction: while comprehensive, it should be more focused. Several paragraphs repeat general knowledge and contain excessive citations; the rationale and specific research hypothesis could be stated more clearly. Response: Thank you very much. We have revised the Introduction according your suggestions in the manuscript.
2. Methods: although detailed, the section would benefit from better organization (e.g., concise subsections, removal of redundant manufacturer details). The statistical approach should specify n, type of post hoc tests, and confidence intervals. Response: Thank you very much. We have revised it according your suggestions in the manuscript. We have supplemented the specify n in the statistical approach; however, in accordance with the journal's requirements, the manufacturers of the reagents used and their respective regions must be specified in the "Materials and Methods" section.
3. Results presentation: figures are numerous and informative, but labeling (e.g., “a–h”) and legends could be simplified for readability. Including a schematic summary of the proposed mechanism would help readers. Response: Thank you very much. We have revised it according your suggestions in the manuscript. And the schema illustrating the mechanism of regulation of chicken ovarian functions by FSH and FOXOs has been added in Figure 7. 4. Discussion: this section is long and partially repetitive of the Results. Please focus on novel findings, limit citations to essential studies, and discuss the potential physiological implications for follicle hierarchy in hens.
5. Mechanistic claims: while the experiments convincingly show PI3K/Akt-mediated phosphorylation of FOXO1, the suggested crosstalk with the p62/Keap1/Nrf2 pathway remains correlative and requires cautious interpretation. The term “completely mediated” or “directly results in” should be softened unless supported by additional functional assays (e.g., pathway inhibition combined with rescue). Response: Thank you for the constructive comment. We agree that the current results support a correlation rather than a direct causal relationship between PI3K/Akt and p62/Keap1/Nrf2 pathways. The overstrong expressions (“completely mediated,” “directly results in”) have been softened, and we have clarified this limitation in the Discussion.
6. Minor comments: ①Gene and protein symbols should be italicized or capitalized according to standard nomenclature. Response: Thank you very much. We have thoroughly revised the manuscript to ensure that all gene and protein symbols follow the standard nomenclature guidelines, with gene symbols italicized and protein symbols presented in regular or capitalized form as appropriate.
②Figures should include scale bars and molecular weight markers for Western blots. Response: Thank you very much. We have revised it. The scale bars of the immunofluorescence images have been marked with red line segments on the figures, and the molecular weights of the proteins have been indicated on the right side of the Western Blot bands.
③The abbreviations list is missing or incomplete. Response: Thank you very much. A comprehensive list of abbreviations has been added to the revised manuscript.
④Some references are outdated or duplicated; please check citation accuracy and formatting according to Cells guidelines. Response: Thank you for your helpful suggestion. We have rechecked all citations and reference entries, corrected formatting issues, removed duplicates, and replaced outdated sources following Cells guidelines. Response to Comments on the Quality of English Language |
|
English and Style: the manuscript requires substantial language polishing to improve clarity, conciseness, and readability. Some sentences are overly long, repetitive, or contain grammatical inconsistencies. Professional English editing is strongly recommended. Response: We appreciate the suggestion regarding language quality. In response, we have thoroughly polished the manuscript to enhance clarity, conciseness, and readability. Grammatical errors and redundancies have been corrected.
|
Round 2
Reviewer 1 Report
Comments and Suggestions for Authors
The performed revision resulted improvement the manuscript. Nevertheless, it still contains several weak points:
ABSTRACT. The protocol of experiments is not properly described. Only part of the used methods experimental and analytical methods has been indicated.
INTRODUCTION. Please explain, why just this model and just these parameters have been selected for analysis.
MATERIALS AND METHODS. Did the author used chicken or porcine FSH? They could have different action on chicken ovarian cells.
2.7. Quantitative real-time PCR analysis „cDNA sequence of P62 was referenced to literature“. Please indicate the source of information concerning sequence of all the primers.
2.10. Statistical analysis. Application of Steden’s t-test is possible, if the normality of distribution has been shown. Did the authors perform normality test?
RESULTS. „P“ is not necessary to indicate in the text, if it was indicated on graphs.
DISCUSSION. Please indicate clearly, what for query the authors try to address in each part of Discussion, and what is novelty of each result presented by the authors in comparizon with their previous publications. The pragraph rule of scientific writing is not always used. „This finding provides new insights into the molecular basis of FSH-mediated regulation of follicular development and identifies potential molecular targets for improving follicle selection efficiency and enhancing laying performance in chicken.“ Please explain, how the results of the author’s study could increase laying performance?